# End-of-Life Management of Biodegradable Plastic Dog Poop Bags through Composting of Green Waste

**DOI:** 10.3390/ma15082869

**Published:** 2022-04-14

**Authors:** Danuta Dróżdż, Krystyna Malińska, Przemysław Postawa, Tomasz Stachowiak, Dorota Nowak

**Affiliations:** 1Faculty of Infrastructure and Environment, Czestochowa University of Technology, Brzeznicka 60A, 42-200 Czestochowa, Poland; dorota.nowak@pcz.pl; 2Faculty of Mechanical Engineering and Computer Science, Czestochowa University of Technology, Armii Krajowej 19c, 42-200 Czestochowa, Poland; przemyslaw.postawa@pcz.pl (P.P.); tomasz.stachowiak@pcz.pl (T.S.)

**Keywords:** compostability, biodegradability, bags for collecting dog poop, compost, plastic waste, end-of-life management

## Abstract

Plastic waste derived from plastic dog poop bags (DPBs) could be considered a negligible source of plastic waste. However, it is estimated that this kind of waste contributes to 0.6% of the total plastic waste generated worldwide, and it is expected to increase in the following years. Plastic dog poop bags can be replaced with biodegradable and bio-based alternatives. These alternatives can biodegrade in various environments such as composting, soil, or water and thus allow for end-of-life management without the risk of contaminating the environment with microplastics. However, not all biodegradable bags are always compostable. In this study, we tested composting as the end-of-life management of selected biodegradable dog poop bags (i.e., prototypes of bags and commercially available bags). We analyzed the biodegradation of selected biodegradable plastic dog poop bags during the composting of green waste in laboratory composting reactors after 4 weeks and 90 days of composting. All the investigated DPBs decomposed 100% after 90 days of composting. However, the fresh compost obtained after the 90-day composting of green waste mixed with the investigated bags containing dog poop did not demonstrate high quality.

## 1. Introduction

Plastic waste has been a major problem worldwide. It has been observed that the global production of plastic waste has increased dramatically over the past 50 years. In 2018, the global production of plastic waste exceeded 350 million tonnes [1,2] while in Europe, it was estimated at 61.8 million tonnes. Asia is the leader in generating plastic waste—51%, followed by North America—18% and Europe—17%. Africa, South America, and Australia contribute to 14% of total plastic waste [3]. Almost 40% of plastics are used for the production of packaging. Polypropylene (PP) and polyethylene (PE-LD) are the most common polymers used to produce food packaging, containers, agricultural mulches, cover films, pipes, single-use bags, etc. [3,4]. In 2018, 9.4 million tonnes of plastic waste were recycled in Europe. One of the most difficult types of waste to manage is plastic packaging. Plastic waste from packaging is estimated at more than 17 million tonnes per year. This kind of waste is mainly recycled (42%), used for energy recovery (39.5%), or landfilled (18.5%). The Czech Republic is the recycling leader among the European countries as it can mechanically recycle 52% of waste residues in the form of plastic packaging [3].

Given these statistics on the global plastic waste generation, it may seem that plastic waste derived from plastic dog poop bags (DPB) could be considered a negligible source of plastic waste [5]. However, it is not. Plastic dog poop bags are single-use bags for collecting dog excrements and are discarded directly after use. The common practice is that the discarded plastic dog poop bags enter the waste stream of municipal waste and are either landfilled or incinerated. The majority of these bags are produced from conventional fossil-derived polymers, which do not biodegrade in the environment and in reality cannot be recycled. According to Mai et al. (2022), dog poop bags contribute to 0.6% of the total plastic waste generation worldwide. In Europe alone, there are over 85 million dogs, and this number is expected to increase in the next few years. It is assumed that an average dog produces about 68 kg/year of excrements, which gives over 5 million tons of dog poop per year [6].

The issue of plastic waste from dog poop bags is particularly problematic in urban areas where dog owners typically use plastic bags to pick up dog poop and throw them into available municipal waste bins or directly into the environment, posing a threat of microplastic contamination. The municipal waste is collected and then landfilled [7]. It is reported that the uncontrolled accumulation of significant amounts of dog poop could lead to the contamination of soil and surface water. It could also pose a threat to living organisms due to the exposure to *Salmonella*, *Escherichia coli*, and *Giardia lamblia* often present in dog poop. This, in turn, could lead to toxocariasis, dipylidosis, intestinal problems, anquilostomiasis, tricuriasis, giardiasis, and cryptosporidiosis in humans [8].

Conventional non-biodegradable fossil-derived plastics used mostly for manufacturing dog poop bags can be replaced with biodegradable plastics derived from biomass sources [9]. It has to be pointed out that biodegradable plastics can be also derived from fossil sources. These biodegradable plastics can degrade in the environment, e.g., in compost, in soil, in water, and by the action of microorganisms [10]. Currently, there are many types of biodegradable dog poop bags that are available on the market. However, not all of them meet the biodegradability and compostability standards. Often the word “biodegradable” itself does not mean that it decomposes quickly and in an environmentally friendly way. In addition, one common perception is that “biodegradable” plastics when discarded into the environment will easily disappear [2,10]. 

Biodegradable plastics in different environments are exposed to the activity of various indigenous microorganisms. First, these plastics break down into smaller components, and after some time, these smaller components are converted under aerobic conditions into CO_2_ or anaerobic into CO_2_ and CH_4_, and new microbial biomass. Biodegradation in open environments depends on a number of factors, in particular: specific environmental conditions (temperature, pH, moisture content, microorganisms, nutrients) and the structure of the polymer (chemical bonds, crystallinity, surface-to-volume ratio, additives, etc.) [11]. The biodegradation of plastic is assessed in laboratory conditions (often as a powder or a testing sample—not a product itself) following the existing industrial standards such as the European EN-13432 or the international ASTM D6400 and ASTM D6954 [4,12]. According to the EN-13432, a “biodegradable” product must be at least 90% decomposed within 6 months, whereas ASTM D6954 indicates that a “biodegradable” product should biodegrade in less than 2 years without leaving any residues that may be harmful to the environment [7]. It has to be pointed out that the determination of biodegradability of various plastic materials following the available standards is carried out in the controlled laboratory conditions, and thus, it does not give the overall information on how biodegradation occurs in different environments [11,12]. Thus, the validation of laboratory-determined biodegradability of plastics in different environments (i.e., compost, soil, water) is recommended. In addition, potential contamination of the environment with microplastics should be assessed [13].

Therefore, there is a need for a different approach to the use of plastic dog poop bags and end-of-life management of plastic waste from dog poop bags. First and foremost, dog owners should be encouraged to use biodegradable plastic dog poop bags that could be easily discarded into special containers. Dog owners should be provided with clearly stated information on the type of dog poop bags and disposal practices. This is of utmost importance as biodegradable plastics cannot be discarded together with recyclable plastics. This is crucial—particularly in urban areas—that dog owners should be provided with a special collection system for biodegradable plastic dog poop bags which in turn allows biological treatment such as composting [5]. However, some biodegradable plastics are not always compostable and do not decompose during composting [9]. In addition, some biodegradable plastics without proper end-of-life management options should not be introduced to the market [14]. Therefore, dog owners should be provided with clearly stated information on the properties of biodegradable dog poop bags and guided on how to dispose of waste from biodegradable dog poop bags.

The overall goal of this paper is to report the results from the study that analyzed selected biodegradable plastic dog poop bags and their end-of-life management by composting with green waste as a sustainable way of handling waste from biodegradable plastic dog poop bags. The scope of the study covers: (1) the production of biodegradable non-fossil-derived plastic dog poop bags, (2) analysis of selected biodegradable plastic dog poop bags available on the market, (3) lab-scale composting of selected biodegradable plastic dog poop bags, (4) analysis of compostability of biodegradable plastic dog poop bags, and (5) the analysis of the composting mixtures prior and after composting.

## 2. Materials and Methods

### 2.1. Materials

#### 2.1.1. Prototypes of Biodegradable Non-Fossil Derived Plastic Bags for Collecting Dog Poop

Two types of dog poop bags (DPB-A and DPB-B) were developed, manufactured, and tested at the Czestochowa University of Technology (Poland). The dog poop bags were produced from innovative three-layered biodegradable non-fossil derived films developed within the Organic PLUS project (Horizon 2020). The three-layered films were manufactured from commercially available Bioplast 400 ELITE (in the outer layers) and Bioplast 400D (in the inner layer) filled with different fillers used in the inner layer. Bioplast 400 ELITE is a fully biodegradable and compostable thermoplastic material without the addition of plasticizers. It contains potato starch and bio-based polymers. It is biodegradable and compostable as it complies with EN 13432 and is certified with “Ok Compost HOME”. Bioplast 400D contains non-GMO natural potato starch without the addition of plasticizers. It also complies with EN 13432. The composition of the films (A and B) manufactured within the Organic PLUS project (Horizon 2020) is presented in Table 1. Both films contained bio-based fillers and pigments. It has to be pointed out that the primary application of the films A and B manufactured in the Organic PLUS project was agriculture (as mulching films and covers). In this study, we are exploring applications of these films other than in agriculture. Therefore, the thickness of the dog poop prototypes is higher than the films used in commercial dog poop bags. This applies in particular to the dog poop bag B with a thickness of 40 microns. This could have an impact on the biodegradation process in the composting reactor. 

The films were manufactured on an industrial scale with the use of a blown film extrusion line at an industrial partner (Marma Polskie Folie, Poland). The line was equipped with two extruders, which were divided into internal and external layers, respectively. The extruder of the internal layer was dosing Bioplast 400D and fillers through the gravimetric feeders, and the extruder of the outer layers was dosing only Bioplast 400 ELIT. The manufactured films were used to produce dog poop bags by cutting and sealing to form a bag with the laboratory stand of a capacitive welder with a weld length of 600 mm and a width of 2 mm. 

#### 2.1.2. Commercially Available Biodegradable Plastic Bags for Collecting Dog Poop

We selected 3 commercially available biodegradable plastic dog poop bags (referred to as DPB-C, DPB-D and DPB-E) for our study. The selection was made based on the availability, the price and the feedback from the dog owners. The manufacturers of these bags did not provide specific information on the type of material they were produced from. All the bags were marketed as biodegradable bags.

#### 2.1.3. Characteristics of the Investigated Biodegradable Plastic Dog Poop Bags

Prior to the composting experiment, the investigated dog poop bags (A, B, C, D and E) were analyzed to determine selected characteristics (i.e., dimensions, color and thickness, composition, etc.). All bags had different dimensions, but their surface area was very similar. The DPB-A and DPB-B had a three-layered structure and thus demonstrated higher thickness, which may have an impact on degradation during the process of composting. The most important properties of the investigated dog poop bags are presented in Table 2. Only the composition of dog poop bags A and B is known. The composition of the commercially available bags (C, D, E) differed. Although some information on the composition was provided by the manufacturers, the composition of the DPB-C, DBP-D and DBP-E bags was additionally determined through the differential scanning calorimetry DSC to confirm the information on the labels (results not presented here).

### 2.2. Methods

#### 2.2.1. Laboratory Composting Set-Up

Three identical laboratory composting reactors with the volume of 60 L each were used for composting. They were equipped with a forced aeration system, oxygen flow controllers, and a set for collecting leachate and condensate [15]. Perforated plates were also placed at the bottom of the reactors, which assured free air circulation at the bottom of the reactor. A detailed description of the composting set-up is reported in the previous work of the authors [16].

#### 2.2.2. Laboratory Composting

Three laboratory composting reactors (indicated throughout the text as R1, R2 and R3) were used for the composting trials (Table 3). The selected dog poop bags DPB A–E (without dog poop) were placed in the reactors R1 and R2 (R1 and R2 were treated as replications for the composting process) with freshly mowed grass (about 12 kg per reactor) and arranged in 5 layers with the depth of the reactor (Figure 1). Each DPB was tested in 5 replications per composting reactor (R1 and R2). In parallel to the composting of the DPBs in R1 and R2, we also composted the investigated DPBs filled in with dog poop and mixed with the mowed grass in the composting reactor R3. The number of layers was reduced to 3 (this resulted in 3 replications for each type of the bag). The composting of dog poop bags in R1 and R2 was run for 4 weeks, and then, the investigated bags were taken out from the reactors for testing. Composting of the bags containing dog poop in R3 was run longer and stopped after 90 days. The flow rate was 35 L/h. Composting dynamics was monitored by daily measurements of temperature in the composting reactors R1, R2 and R3. During composting, we also collected leachate from all composting reactors (the results are not presented here). 

#### 2.2.3. Physical, Chemical and Microbiological Analysis of the Composted Mixtures

Freshly mowed grass and the composted mixtures were tested for moisture content (MC), organic matter (OM), total nitrogen (TN), phosphorus (P), total organic carbon (TOC), pH and temperature. MC was determined by drying the samples at 105 °C to constant weight. The content of organic matter (OM) was determined by the loss on ignition of the dried mass in a muffle furnace at 550 °C for 4 h. Total nitrogen (TN) was analyzed by the Kjeldahl method. The content of phosphorus was determined by the spectrophotometric method (Hach Lange DR 5000, HACH LANGE Sp. z o. o., Wrocław, Poland) with ammonium molybdate. The content of total organic carbon (TOC) was analyzed with the organic carbon analyzer (TOC) (Carbon Analyzer Multi N/C 2100, Analytikjena, Jena, Germany) at temperature of about 1200 °C with oxidation in the stream synthetic air. The pH was measured with a laboratory pH multifunction device/conductometer (Elmetron CPC-505, ELMETRON Sp.j., Zabrze, Poland). All samples were tested in 3 replications. 

The fresh compost obtained from mowed grass and mixed with dog poop bags containing dog poop was sampled from the composting reactor R3 after 90 days of composting. The 5 compost samples were taken from the compost batch to assure representativeness and were individually tested for *Enterococcus faecalis*, *Coliform* and *Salmonella*. In the case of *Enterococcus* and coliform bacteria, 10 g of the compost was shaken in 90 mL of physiological saline for 15 min, and then, a series of 10-fold dilutions were made in the range 10^−2^–10^−6^ for composted mixtures. The prepared sample was spread on the appropriate identification media.

Determination of coliform index was performed with the test on the Eijkman lactose growth medium (LPB) (BTL, Warszawa, Poland), which was inoculated with the prepared dilutions. The cultures were grown at 37 °C for 24 h. The presence of coliform bacteria was determined based on the change in color from purple to yellow of the growth media due to fermentation of lactose and formation of gas. Identification of *Escherichia coli* was performed with the use of the Mc Conkey growth medium. The cultures were incubated at 44 °C for 24 h. *Escherichia coli* grew in the form of red colonies and did not change the color of the growth medium. To detect fecal streptococci (*Enterococcus faecalis*), the test-tube method was used with broth containing sodium azide and bromocresol purple (APB). Sodium azide and bromocresol purple medium were used for preliminary tests and for tests confirming the medium with ethyl violet (AFP). The cultures prepared in this way were incubated at 37 °C for 24–48 h. To isolate the bacteria of the genus *Salmonella*, the preliminary multiplication in buffered peptone water was performed, and the bacteria was identified on a very strongly selective medium (SS Agar medium), which allows this type of bacteria to grow in the form of yellow colonies with a characteristic black center. The cultures were incubated at 37 °C for 24 h. All inoculations were performed in 3 parallel replications. The number of *Enterococcus faecalis* and coliform bacteria was expressed as the colony-forming unit (CFU) of bacteria per 1 g of material and in the form of titer, i.e., the smallest volume of the tested material in which the presence of the determined bacteria is still found. Bacteria of the genus *Salmonella*—their presence or absence—was confirmed in the tested material.

#### 2.2.4. Determination of Mass Loss and Structural Changes of DPBs after Composting

Prior to composting, the selected DPBs were subject to weighing, thickness assessment, and the analysis of selected properties. We determined the weight of each bag and placed them in the composting reactors. The mass loss is the indication of the degradation processes that occurred during composting. The weight of the bags prior to and after composting was determined with the use of a Sartorius microanalytical balance (the accuracy of 0.01 mg). Before and after composting, all the bags were photographed. After 4 weeks of composting, the bags were carefully taken out of the composting reactors, washed from dirt, and air-dried. Next, they were weighed, and then, their structure was visually analyzed for changes in shape, color, etc.

#### 2.2.5. Statistical Analysis

Standard deviation was calculated for the results from laboratory tests and the mass losses of the investigated bags during composting. 

## 3. Results and Discussion

### 3.1. Laboratory Composting

#### 3.1.1. Composting Dynamics

The observed temperature evolution during composting was typical for this process. At the beginning of the process, the temperature increase was dynamic, which was related to the active phase of composting [16,17]. The highest temperature was recorded on the second day in the composting reactor R2, and it exceeded 66 °C. In other reactors, the temperature in the first days exceeded 63 °C. High temperature (50–70 °C) during the composting process allows hygienization of the compost and biological stability. 

Composting dynamics were monitored by daily temperature measurements (Figure 2). 

#### 3.1.2. Physical, Chemical and Microbiological Properties of the Composted Mixtures and Fresh Compost

The composted mixture consisted of freshly mowed grass which contained 82.59% of moisture content, 83.62% of organic matter, 2.86% of N, 46.45% of C org, 7.42 mg·g^−1^ of P_2_O_5_, and demonstrated pH of 6.57, C/N 16.24. After 4 weeks of composting (the active phase), all composted mixtures (from the composting reactors R1 and R2) and fresh compost (from the composting reactor R3 after 90 days of composting) were analyzed, and the results are presented in Table 4. 

During the process of composting, the mass of the composting mixtures changed. The mass loss in R1 and R2 was 33% and 42%, respectively. In the case of R3, it was 46%. The loss of mass was due to longer composting (90 days). The weight loss during composting can vary from 10 to almost 60% depending on the used substrates, the content of organic matter and the temperature of the compost mixture [18]. The suitable moisture content of 60% facilitates microbial activity during composting. The minimum threshold for sufficient activity during the composting process for microorganisms is 50% of moisture content. On the other hand, the level of 60–70% ensures maximum work activity for microorganisms [19,20].

According to the data presented in Table 4, all composted mixtures demonstrated similar characteristics, i.e., the content of organic matter, moisture content, nitrogen, carbon, and phosphorus. They showed high content of organic matter and increased content of phosphorus.

The high pH is related to the substrate used in the composting process as grass naturally raises the pH of composted mixtures. Composting mixtures were characterized by low C/N, which was initially low. In addition, a low C/N ratio can be related to the dynamic loss of C and N and CO_2_ emission during the composting process [21].

In addition, the fresh compost (sampled after 90 days) that contained DPBs with the dog poop (R3) was subject to microbiological analysis, including *Salmonella*, coliform and *Enterococcus faecalis* bacteria. The microbiological analysis was carried out in 5 replications (Table 5). 

According to the European Commission Regulation No. 142/2011, the content of *Enterococcus faecalis* should not exceed 5000 CFU per g of material. In addition, bacteria of the genus *Salmonella* should not be present in the compost. Comparing the results with the legal regulations, the compost from the grass with decomposed bags containing dog poop is microbiologically safe [22,23]. However, the fresh compost contained *coli* bacteria. This was due to high moisture content, which could facilitate the growth of *coli* bacteria and the fact that this fresh compost was sampled prior to the maturation phase [24]. 

### 3.2. Characteristics of the Investigated Dog Poop Bags after Composting

#### Mass Loss after the 4-Week Laboratory Composting 

After 4 weeks of composting, the investigated DPBs were taken out from the composting reactors R1 and R2. After the removal, the bags were washed in running water to remove grass residues and then dried in laboratory conditions of 25 °C for 48 h. Due to the advanced decomposition processes, no laboratory dryer was used. Dried bags were characterized visually for changes in volume, color and size of the bags after composting. The changes in the structure of the investigated DPBs are presented in Table 6. 

Next, all bags were weighed. The mass loss was determined from the weights recorded prior to and after composting. The average mass losses of the investigated DPBs from the composting reactors R1 and R2 are presented in Figure 3 and Figure 4, respectively. The average mass loss of each bag was determined from all the layers in the composting reactors. 

Degradation of the investigated DPBs during composting is a complex phenomenon and depends on various factors. The most important factors include the type of biodegradable plastic material (i.e., type of a polymer, physical and thermal properties of the material, size and thickness of a product, etc.), the composition of the composting mixture (i.e., moisture content, pH, organic matter, indigenous microorganisms, oxygen availability, C/N ratio, etc.) and composting system (i.e., composting windrows, composting reactors, static vs. turned windrows, industrial vs. home composting) and the process parameters (i.e., air flow rate, mixing, temperature). In our study, we observed differences in the mass loss after 4 weeks of composting among the investigated bags, which were arranged in different layers in the composting reactors (Figure 5 and Figure 6). 

Our observations confirmed that during the process of composting, the biodegradable plastic bags underwent degradation slightly differently, depending on the depth in the composting mixture. This is due to several aspects including the fact that the composted mixture (depending on the type and the ratio of substrates in a mixture) is often very heterogeneous. The temperature during composting is the highest in the inner parts in the reactor. The outer parts in the reactor can contain less moisture, whereas moisture at the bottom of the composted mixture could be higher. High moisture content in the layers of the composted mixture can prevent proper airflow through the composting matrix. This, in turn, can limit the oxygen availability to the microorganisms during composting. Therefore, the composting of biodegradable plastics should be optimized for efficient end-of-life management. All these factors should be taken into account when the industrial composting of biodegradable plastics as end-of-life management is considered. This is of particular importance, especially at the composting facilities that perform composting of biodegradable plastics mixed with organic waste. In reality, these facilities need to adjust the process to manage biodegradable plastics in an efficient manner, as in most cases, these facilities were not designed to process these types of materials [25].

Dog poop bags that are biodegradable are not always compostable. In our study, all the investigated bags were biodegradable and compostable. This was evidenced by the total decomposition of the DPBs containing dog poop during the 90 days composting in the composting reactor R3. The DPB-A and DPB-B that were produced from the innovative biodegradable films intended for agricultural applications (developed within the *Organic+* project) proved to be suitable for collecting dog poop and be end-of-life managed by composting. The obtained results demonstrated that these bags are compostable and decompose entirely within 90 days of composting with green waste. This, however, could be a feasible option for managing biodegradable waste if an adequate collection system for waste from biodegradable plastic dog poop bags is available. 

## 4. Summary and Conclusions

Single-use biodegradable dog poop plastic bags can be alternatives to conventional non-biodegradable plastic bags. These alternatives—if discarded with organic waste—can be easily decomposed during industrial composting. However, without a proper waste collection system for biodegradable dog poop bags—especially in urban areas—and sustainable end-of-life management practices, dog owners encouraged to use these biodegradable alternatives are most likely to discard these bags together with municipal solid waste. Therefore, it seems that using biodegradable dog poop bags is justified if this type of waste can be collected with organic waste and treated with biological methods, e.g., composting. 

Based on the results obtained from this study, the following conclusions were formulated:After 4 weeks of laboratory composting, the average mass losses for DPBs A, B, C, D and E from R1 and R2 were as follows: 52%, 49%, 73%, 73%, 12% and 66%, 40%, 50%, 71%, and 11%, respectively.Although the investigated DPBs A–E after 4 weeks of laboratory composting demonstrated differences in decomposition (i.e., quantitative as well qualitative changes), all of them decomposed 100% after 90 days of laboratory composting.The investigated DPBs were arranged in layers at different depths inside the composting reactors, and thus could be exposed to different composting conditions related to oxygen concentration, moisture content, heat and air exchange, etc. This could have an effect on the degradation of these bags, which was evidenced by the reported mass losses after 4 weeks of composting.The fresh compost obtained after 90 days of laboratory composting of mowed grass mixed with bags containing dog poop showed insufficient quality, mostly in terms of microbial stability. However, after maturation, the obtained compost could be used for, e.g., soil reclamation.

Future work will include the study on composting as the end-of-life management of discarded biodegradable dog poop bags mixed with selected organic waste, e.g., kitchen and backyard waste in composting windrows and home composting bins to dispose of discarded dog poop bags and to convert organic waste into mature compost for soil applications. 

## Figures and Tables

**Figure 1 materials-15-02869-f001:**
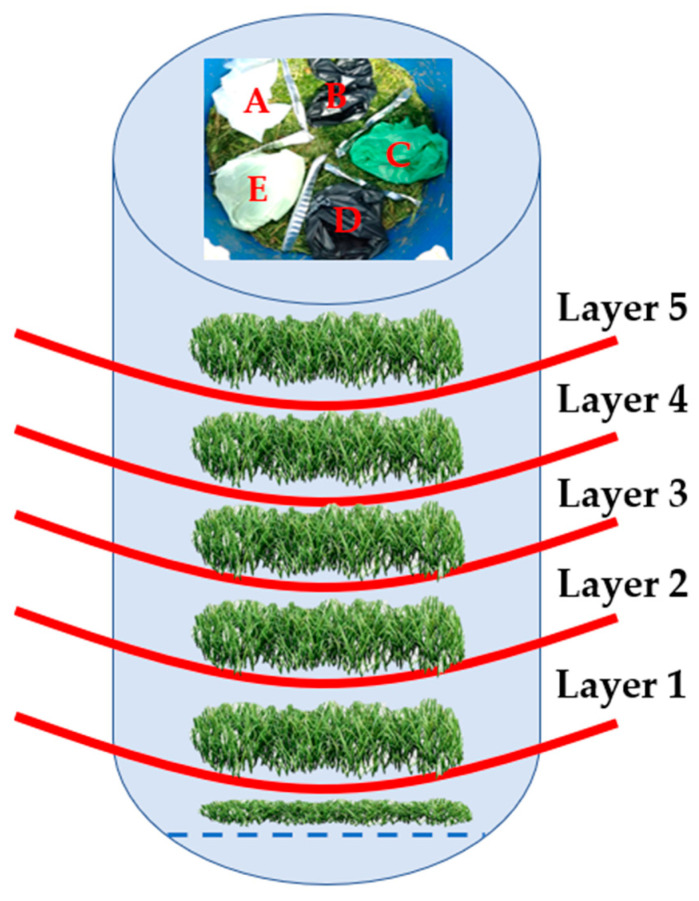
The arrangement of the investigated DPBs (A–E) in the composting reactors R1 and R2.

**Figure 2 materials-15-02869-f002:**
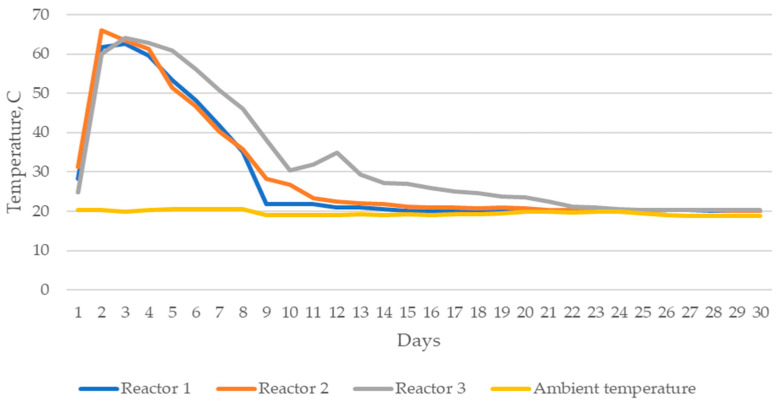
Temperature evolution in the composting reactors R1, R2 and R3 was recorded for a period of 4 weeks.

**Figure 3 materials-15-02869-f003:**
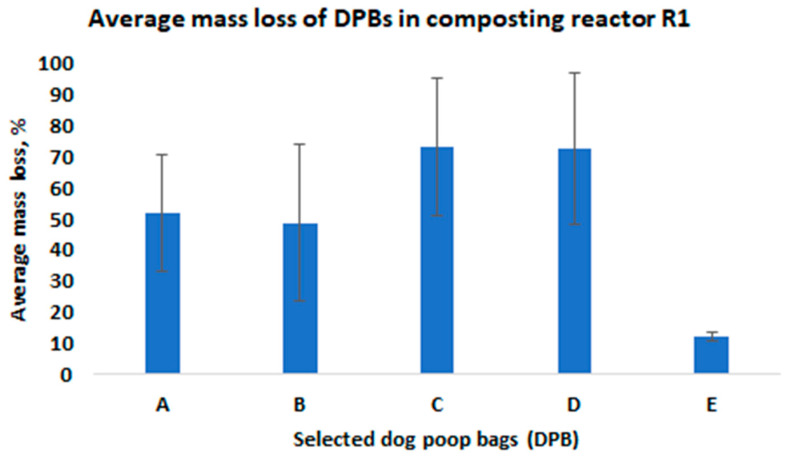
The average mass loss of the investigated DPBs in the composting reactor R1 after 4 weeks of composting.

**Figure 4 materials-15-02869-f004:**
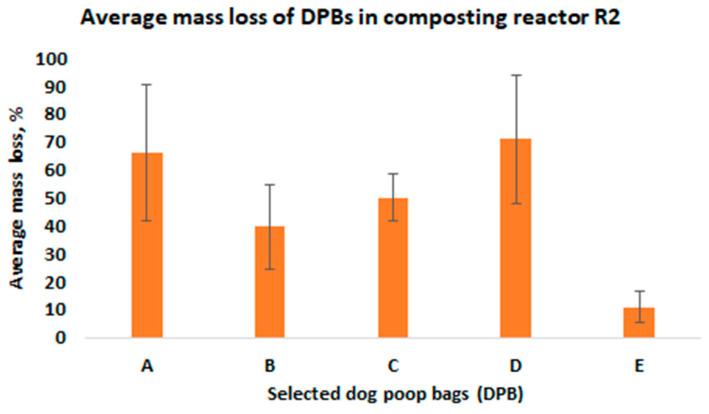
The average mass loss of the investigated DPBs in the composting reactor R2 after 4 weeks of composting.

**Figure 5 materials-15-02869-f005:**
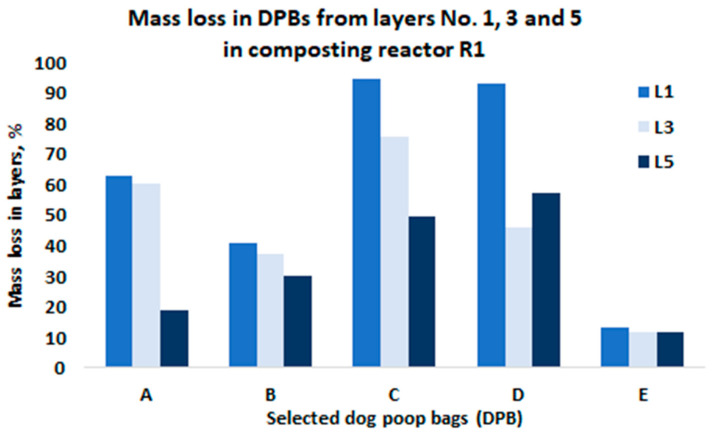
Mass loss of the investigated dog poop bags sampled from the layers in the composting reactor R1.

**Figure 6 materials-15-02869-f006:**
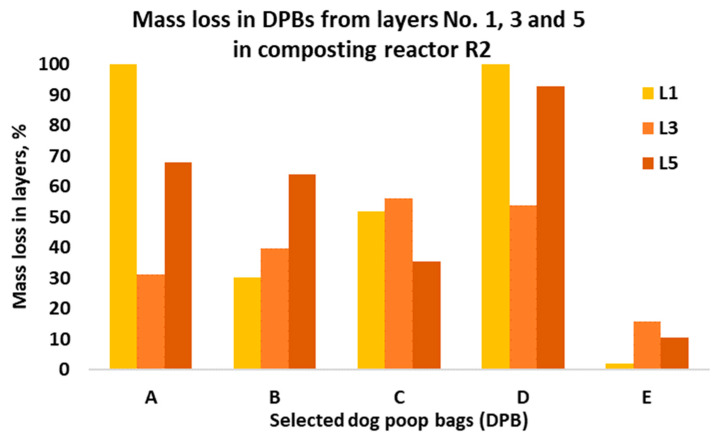
Mass loss of the investigated dog poop bags sampled from the layers in the composting reactor R2.

**Table 1 materials-15-02869-t001:** Biodegradable non-fossil derived plastic films A and B, which were used to manufacture dog poop bags (DPB-A and DPB-B).

Type of DPB	DPB-A	DPB-B
Thickness	25 microns	40 microns
Material	Bioplast 400D	Bioplast 400D
	Bioplast 400 ELIT	Bioplast 400 ELIT
Filler used in the inner layer	20% of CaCO_3_ bio-based from seashells no pigment	5% of black bio-based pigment
Color	semi-transparent	black

**Table 2 materials-15-02869-t002:** Characteristics of the dog poop bags (DPB) used in the experiment.

DPBs	Visualization	Properties	Composition
A	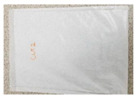	Width: 150 mm Height: 210 mm Thickness: 0.025 mm Color: White, milky Non-commercial product Average weight: 2.65 g	Biodegradable with calcium carbonate as a filler. Origin: Non-fossil derived.
B	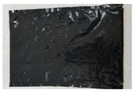	Width: 170 mm Height: 245 mm Thickness: 0.04 mm Color: Black, glossy Non-commercial product Average weight: 3.74 g	Biodegradable with organic pigment. Origin: Non-fossil derived.
C	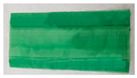	Width: 145 mm Height: 295 mm Thickness: 0.017 mm Color: Green, matte Commercially available product Average weight: 2.69 g	A dog poop bag made of biodegradable plastics *. Origin: No data on the origin.
D	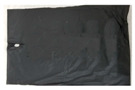	Width: 140 mm Height: 310 mm Thickness: 0.02 mm Color: Graphite, matte Commercially available product Average weight: 2.86 g	A dog poop bag produced from corn starch *. Origin: Bio-based.
E	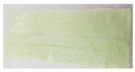	Width: 140 mm Height: 310 mm Thickness: 0.02 mm Color: Light yellow, matte Commercially available product Average weight: 2.608 g	A dog poop bag produced from corn starch *. Origin: Bio-based.

* Based on the information from the manufacturers of the commercially available bags (either from the labels or the manufacturer’s website).

**Table 3 materials-15-02869-t003:** The arrangement of the investigated DPBs in the composting reactors.

DPB	Reactor R1	Reactor R2	Reactor R3
	(5 Layers)	(5 Layers)	(3 Layers)
**A**	1 bag per each layer	1 bag per each layer	1 bag with 20 g of dog poop per each layer
**B**	1 bag per each layer	1 bag per each layer	1 bag with 20 g of dog poop per each layer
**C**	1 bag per each layer	1 bag per each layer	1 bag with 20 g of dog poop per each layer
**D**	1 bag per each layer	1 bag per each layer	1 bag with 20 g of dog poop per each layer
**E**	1 bag per each layer	1 bag per each layer	1 bag with 20 g of dog poop per each layer

**Table 4 materials-15-02869-t004:** Selected characteristics of the composted mixtures after 4 weeks (R1, R2) and 90 days (R3) of composting.

Composted Mixtures	MC	OM	pH	EC	Total N	P_2_O_5_	TOC	Ratio C/N
	%	%	-	mS	%	mg·g^−1^	%	-
**Composted mixture (R1)**	85.14 ± 1.31	72.69 ± 3.24	8.877	5.099	3.98 ± 0.08	10.76 ± 0.07	40.38 ± 0.02	10.14
**Composted mixture (R2)**	85.17 ± 1.50	71.45 ± 4.11	8.866	3.992	4.70 ± 0.08	10.62 ± 0.07	39.69 ± 0.05	8.44
**Fresh** **compost (R3)**	86.12 ± 1.32	70.39 ± 3.18	8.897	3.893	4.52 ± 0.07	10.49 ± 0.06	38.72 ± 0.03	8.56

MC—moisture content, OM—organic matter, EC—electrolytic conductivity, TOC—total organic carbon.

**Table 5 materials-15-02869-t005:** The content of pathogenic microorganisms in the fresh compost R3.

Replication Number	* CFU *Enterococcus faecalis*/g Material	*Enterococcus faecalis* Index	CFU	Coliform Index	Bacteria of the Genus *Salmonella*
			*Escherichia coli*/g Material		
Sample 1	1.5 × 10^2^	10^−2^	15 × 10^5^	10^−6^	Not isolated
Sample 2	2.4 × 10^2^	10^−2^	24 × 10^5^	10^−7^	Not isolated
Sample 3	1.4 × 10^2^	10^−3^	15 × 10^5^	10^−7^	Not isolated
Sample 4	46 × 10^2^	10^−2^	46 × 10^5^	10^−7^	Not isolated
Sample 5	21 × 10^2^	10^−2^	21 × 10^5^	10^−7^	Not isolated

* CFU—a colony-forming unit.

**Table 6 materials-15-02869-t006:** Characteristics of DPBs after the 4-week composting.

DPB	Before and after Composting (Layer 1 and Layer 5)	Changes in DPBs Observed after Composting
A	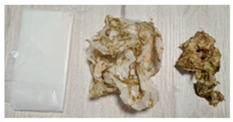	Volume reduction. Color changes. Delicate, falling apart.
B	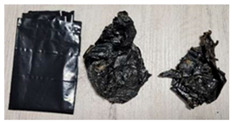	Moderate volume reduction.
C	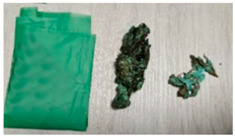	Significant change in mass and volume. Change in hardness.
D	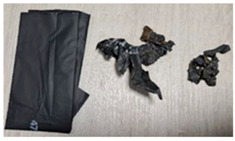	Volume reduction. Delicate, falling apart.
E	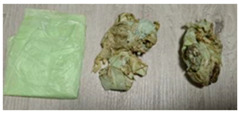	Volume reduction. Change in hardness.

## Data Availability

Not applicable.

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
