# Peer review of "End-of-Life Management of Biodegradable Plastic Dog Poop Bags through Composting of Green Waste"

_materials, 2022, doi:10.3390/ma15082869_

Round 1

Reviewer 1 Report

This article introduces the composting behaviors of dog poop bags (DGB) made of biodegradable plastics in some composts. The experiments were systematically organized and performed. The DGBs were demonstrated to be compostable under the applied conditions. The DGBs with dog poop were also examined. I think that this article is acceptable for the publication in materials.

1) The authors should give more information on the Bioplast 400 ELITE and D. Although it is stated “It contains potato starch and bio-based polymers.” (page 3, line 10-11 from the bottom), the kinds and the contents of the bio-based polymers should be specified.

2) What is “EC” in Table 4? The abbreviations should be clearly defined. The method to determine C org should be described in the experimental section.

3) I recommend to the authors to discuss the biodegradation behaviors of the DGBs in relation to the kinds of the materials.

4) Some experimental conditions are not clear. How are the amounts of dog poop in the DGBs in R3?

Author Response

Thank you so much for your comments. We revised the manuscript taking your comments into considerations to improve it.

Regarding the comment #1: We provided all the information on these polymers we received from the manufacturer. These 2 polymers are commercially available, not produced by us, so we are limited to the data we received from the manufacturer. It has to be pointed out that we also received confirmation on the fact that these are not GMO potato and there are no plasticizers.

Regarding the comment #2: We agree, all acronyms and abbreviations should be explained. Thank you so much for pointing this out. We also added a sentence explaining determination of Corg.

Regarding the comment #3: We provided some information on biodegradation in relation to our study (pp. 11,12,13) but also added some information as reccommended.

Regarding the comment #4: The table 3 gives all the information about the type and number of the bags we tested along with quantities of dog poop tested in the bags in R3.

Reviewer 2 Report

Overall writing and presentation of the manuscript is poor and thus need careful modifications to improve its quality. The specific comments are as follows.

  1. Three different reactors were designed for the biodegradation of dog poop bags. What is the difference about biodegradation effects? Which one is preferred?
  2. Page 1, paragraph 1, line 4-5: Reference is needed for the statistics givens for the subcontinents.
  3. Page 2, paragraph 1, line 3: Reframe the sentence “we could not…it”.
  4. Page 2, paragraph 2, line 7: The diversity of bacteria must be more than those written. Please check the statement. Coli should be Escherichia coli.
  5. Page 2, paragraph 3, line 6: Write dog poop collecting bags rather than dog bags.
  6. Add one basic background about the sustainable management of plastic dog poop bags in the introduction section.
  7. Page 3, paragraph 1, line 1-9: This section seems a bit like conclusion and future aspect. Please consider rewriting.
  8. Table 2, add a space between number and gram in the column of Properties.
  9. Page 6: some detailed information should be present. How was the material structure analyzed? The design of the three reactors R1, R2, and R3 should be clearly described.
  10. Table 3: What is the difference between the reactor R1 and R2? Figure 5 and Figure 6, mass loss of L3 exhibited somewhat opposite trend for reactor 1 and 2. What is the reason?
  11. Figure 2: what conclusion did the author get based on the recorded temperatures for the 3 reactors? Is the temperature closely related with the biodegradation of tested bags?
  12. Page 7, paragraph 3, line 4: Please mention the appropriate media name for coliforms and others.
  13. Page 7: At the end of material and methods, appropriate statistical analysis process needs to be written.
  14. Page 9: Discussion on the organisms count need to be more elaborative.
  15. Figure 5 and 6: Why there are no error bars?
  16. Page 13: It is better not to use bullets in the conclusion.

Author Response

Thank you so much for your comments.

Regarding the comment #1: The composting reactors (R1,R2,R3) used in this study were identical, as described. R1 and R2 were treated as replications (as indicated in the text) whereas R3 was used to test the same bags but with dog poop. The differences between R1 and R2 (which were treated as replications) in biodegradation of the investigated bags are presented in Fig. 3 and 4. We monitored composting dynamics by measuring daily temperature for all composting reactors which is presented in Fig. 2. We are not sure if we understand the Reviewer's questions.

Regarding the comment #2: Thank you for pointing this out. We revised it.

Regarding the comment #3: We reframed the sentences as reccommended.

Regarding the comment #4: As for the diversity of bacteria in dog poop - we made the reference to the article which provided the examples of the most hazardous - as we also explained in this manuscript. Also, we included the full name for E.coli in the text. Thank you for pointing this out.

Regarding the comment #5: Througout the text we tried to be consistent and use "dog poop bags". We added "poop" where it was missing.

Regarding the comment #6: The introduction already contains the information on how plastic dog poop bags are managed and why it is a problem and why we need alternatives (i.e. biodegradable dog poop bags) supported with appropriate end-of-life managment. We are not sure what the Reviewer means by "sustainable management of plastic dop poop bags".

Regarding the comment #7: In the introduction we gave the overview of the problem and reasons why we are addressing this problem. We are justyfing that using biodegradable plastic dog poop bags is only under certain circumstances. We believe that this section gives sufficient explanation and justification for the work we performed. We decided not to make any changes to this section.

Regarding the comment #8: Thank you for pointing this out. We added the space.

Regarding the comment #9: We did not present the analysis of the structure in this manuscript, and thus we did not describe how the structure of the dog poop bags were analyzed. As for the composting reactors (R1, R2, R3) they were identical and described in the section 2.2.1 with the reference to our previous work where we described the composting set up in more detail.

Regarding the comment #10: There is no difference between the reactor R1 and R2. These were - as indicated in the text - 2 replications. The Fig. 5 and 6 demonstrate differences between degradation of the samples in 3 layers for each of the reactor R 1 and R2. With the depth of the composting bed conditions for biodegradation change and affect biodegradation of the bags. Also, differences between replications can result from the heterogenity of the composting mixture.

Regarding the comment #11: Based on the temperature evolution during composting in all composting reactors we concluded that composting was run properly allowing the favourable conditions for biodegradation of the investigatd bags.

Regarding the comment #12: We indcluded this information. Thank you.

Regarding the comment #13: We added the information on statistical analysis. Thank you.

Regarding the comment #14: Thank you for this comment. Our intention was to present the data on microorganisms in the aspect of end-of-life management and potential use of the obtained compost, not to do and provide a comprehensive analysis. This is what we will do in our next study. 

Regarding the comment #15: The investigated bags were arranged in layers as indicated in the Fig. 1 (in both composting reactors R1 and R2). In each layer we placed 5 different bags per composting reactor, and thus we did not have replications on a layer level. Composting mixtures are very heterogenious. In windrow composting or in composting reactors biodegradable plastics are placed on different heights (i.e. layers). This may have an impact on the biodegradation.

Regarding the comment #16: We are not sure if we understand this comment. We included 4 (numbered) conclusions from our research study. We numbered them for the clarity and convenience to the reader.

Reviewer 3 Report

The authors described the end-of-life composting of biodegradable dog poop plastic bags. The reviewer believes that the scope of this study should be broaden to biodegradable plastics in general to make this study appropriate for publication in Materials. The authors should test the composting method presented in this study to other biodegradable plastic materials, since dog poop plastic bags can be considered a very small portion of plastics being used, thus narrowing the scope of this study. The manuscript can be considered published in Materials if the scope of the study can be at least broadened to "biodegradable plastic bags".

Author Response

Thank you so much for the comment. Our intention was to draw the attantion to the growing problem with managing dog poop bags, especially in urban areas. That is why we focused on the dog poop bags not e.g. single use biodegradable plastic shopping bags or else. According to the statistics - as indicated in the literature - the waste generated from plastic dog poop bags is about 0.6% of the global plastic waste. And this is expected to grow in next years. It is not negligable at all and we believe that this is worth addressing.

In addition, as indicated in this manuscript, the biodegegradable A and B bags were produced from film that was intended for agricultural use (as a mulching/covering film). In this work we tested different application of this film which is using it as a dog poop bag and end-of-life managing through composting. Also, there are differences in many aspects when it comes to handling waste from biodegradable plastics dog poop bags and from e.g. single use biodegradable shopping bags. In our study we wanted to point out these aspects. In addition, the biodegradable plastics dog poop bags issue has not been practically addressed in the literature until 2022. That is why we prefer to keep original approach to this study.

Round 2

Reviewer 2 Report

Generally, I am satisfied with the current version.

Author Response

Thank you so much for all the valuable feedback that helped us to improve our manuscript. In the revised version we also made some langauage improvements.

Reviewer 3 Report

The manuscript can now be accepted by Materials after properly addressing the reviewer's concern.

Author Response

Thank you so much for all the valuable feedback that helped us to improve our manuscript. In the revised version we also made some language improvements.